# Effective Selection for Lower Mortality in Organic Pigs through Selection for Total Number Born and Number of Dead Piglets

**DOI:** 10.3390/ani12141796

**Published:** 2022-07-13

**Authors:** Roos M. Zaalberg, Trine M. Villumsen, Just Jensen, Thinh T. Chu

**Affiliations:** 1Center for Quantitative Genetics and Genomics, Aarhus University, 8830 Tjele, Denmark; tmv@qgg.au.dk (T.M.V.); just.jensen@qgg.au.dk (J.J.); chu.thinh@qgg.au.dk (T.T.C.); 2Faculty of Animal Science, Vietnam National University of Agriculture, Trâu Quỳ, Hanoi 131000, Vietnam

**Keywords:** index selection, restricted gain, organic pig, breeding for welfare

## Abstract

**Simple Summary:**

Breeders use breeding goals to guide genetic gain in a population in a desired direction. Breeding goals consist of economically interesting traits, in which each trait receives an economic value. For example, to increase the size of a piglet litter, breeders use a breeding goal that includes the trait “number of live piglets in a litter” for a specific day after birth. While the litter size is selected using the trait “number of live piglets,” it is composed of two traits: “total number born” and “number of dead piglets.” The current study used simulations to illustrate that selection for litter size could be improved by selecting for the latter two traits rather than the former. This approach corrects for the fact that these two traits are genetically related to each other, but they also have genetic differences. Further, splitting one trait into two traits allows breeders to focus on the specific elements of a trait. For example, organic pig breeders could select for better piglet welfare by splitting “number of live piglets” into two traits, giving a negative economic value to the number of dead piglets.

**Abstract:**

Selection for the number of living pigs on day 11 (L11) aims to reduce piglet mortality and increase litter size simultaneously. This approach could be sub-optimal, especially for organic pig breeding. This study evaluated the effect of selecting for a trait by separating it into two traits. Genetic parameters for L11, the total number born (TNB), and the number of dead piglets at day 11 (D11) were estimated using data obtained from an organic pig population in Denmark. Based on these estimates, two alternative breeding schemes were simulated. Specifically, selection was made using: (1) a breeding goal with L11 only versus (2) a breeding goal with TNB and D11. Different weightings for TNB and D11 were tested. The simulations showed that selection using the first breeding scheme (L11) produced lower annual genetic gain (0.201) compared to the second (TNB and D11; 0.207). A sensitivity analysis showed that the second scheme performed better because it exploited differences in heritability, and accounted for genetic correlations between the two traits. When the second breeding scheme placed more emphasis on D11, D11 declined, whereas genetic gain for L11 remained high (0.190). In conclusion, selection for L11 could be optimized by separating it into two correlated traits with different heritability, reducing piglet mortality and enhancing L11.

## 1. Introduction

The focus of the pig industry on increased litter size and improved growth rate has noticeably increased litter size, but has also increased the average number of dead piglets per litter, termed “piglet mortality” [1]. Piglet mortality is a major welfare issue and an economic challenge for pig farmers across production systems globally. Organic and outdoor pig production tend to have higher piglet mortality rates compared to conventional production, with pre-weaning mortality rates ranging from 18.5% to 39.9% in organic and outdoor herds [2,3,4]. The most common causes of early piglet mortality are stillbirth, crushing by the sow, and starvation or hypothermia due to low birth weight [4,5]. This welfare issue is problematic for organic pig production, especially because it conflicts with the fundamental principles of organic farming, which aims to safeguard the health and welfare of animals [6]. The number of dead piglets in a litter can be reduced through certain management approaches, such as sow confinement, cross-fostering, and close surveillance at birth [1,5]. However, selective breeding could also help reduce piglet mortality to counter certain traits.

Breeding for reduced piglet mortality could be achieved by selecting for sows that are more responsive, potentially reducing the chances of crushing; however, heritability estimates for sow behavior are low [5,7]. Alternatively, reducing the chance of piglet starvation could be achieved by selecting for sows that have enough teats and that have litters with homogenous weight at birth [1,5,8]. Another proven method to reduce piglet mortality is selecting for the number of living piglets after five days, with this trait combining both litter size and piglet mortality [9]. This trait, until recently, has been used by pig breeding programs in Denmark, and has been shown to be successful in reducing, or halting, the increase in piglet mortality in conventional herds [10,11]. 

Yet, concerns existregarding using an approach with a combined trait for both piglet mortality and litter size. Using one trait that selects for litter size and fewer dead piglets overlooks the facts that that these traits are correlated, have different heritability, and are not economically equivalent. There is a genetic correlation between the total number born and the number of dead piglets in a litter; yet, their heritability is not the same [12]. Furthermore, because organic farming principles promote animal wellbeing/welfare and aim to reduce suffering [6], a dead piglet should have a negative economic value. Thus, by separating this combined trait for the number of living piglets [9] into distinct mortality and litter size traits, breeders gain the option to compare the economic and ethical value of living versus dead piglets. Especially for organic pig breeding, it is desirable to include a welfare trait, such as the number of dead piglets in a litter, in breeding goals directly [13]. Selection for sows that have fewer dead piglets in the first 10 days after birth could be an effective approach, as most piglets die during this early period [12,14,15,16]. 

This study aimed to evaluate how genetic gain for a combined trait could be improved by separating it into two traits. Specifically, we separated the “number of living piglets” into “total number born” and “number of dead piglets”.

## 2. Materials and Methods

### 2.1. Simulated Traits

Three sow traits were simulated: the number of living piglets at day 11 after birth (L11), the total number born, including live and stillborn piglets (TNB), and the number of dead piglets at 11 days after birth (D11). The trait L11 = TNB − D11 is similar to the recent combined survival and litter size trait that is used in pig breeding programs in Denmark, namely, the number of living piglets at day 5 (L5) [1,17]. For organic production systems, L11 is a more suitable choice than L5 because (1) piglets are kept inside the farrowing hut and within the fenced area with the sow until day 10; (2) some dead piglets might not be recorded until the fence is removed at day 10; (3) most piglets die during the first 10 days of life [12,14,16]; and (4) survival until day 11 is mainly influenced by the sow’s genotype [12].

### 2.2. Data Collection

Data on TNB and D11 were used to estimate the parameters used in the simulations. Data were collected from March 2018 to October 2021 at an organic multiplier farm in Denmark, named Bovbjerg Økologi A/S. 

On the farm, after the sows were inseminated, they were kept in sow groups containing sows of the same breed/combination. Close to parturition, sows were then transferred to separate huts in a sow field, with individual fields containing 30 to 76 sows. Each sow had a fenced field that included a hut, a feeding and watering station, and a mud pool. Sows in the same sow field typically gave birth within a three-week period. All piglets born to sows in the same field were weaned at the same time. Sows gave birth inside the hut, which was provided with a low fence (“fender”) to keep piglets inside during the first 10 days, whereas the sow was able to go outdoors freely. The trait TNB was recorded immediately after birth by counting all born piglets (dead + alive). The trait D11 was recorded by counting living piglets on the day the fender was removed (day 9, 10, or 11) and subtracting this number from TNB.

Litter size records were recorded for 1235 sows, which had a total of 3977 litters. Of all sows, 781 were Landrace (LLLL) and 466 were Yorkshire boar x LLLL sow (YYLL). Of all litters, 253 were of the breed LLLL, 1935 YYLL, and 1822 Duroc boar x YYLL sow (DDYL). For all litters, both TNB and D11 were recorded. 

### 2.3. Parameter Estimation

Parameters were estimated using DMU software [18]. The parameter estimates used for simulations are shown in Table 1. These estimates were obtained by analyzing the data on TNB and D11 using a bivariate animal model, with the same sub-model used for both traits:
(1)trait=LitterBreed+Parity+HerdYearSeason+IDsow+PEsow+e
where trait was TNB or D11. Fixed effects were LitterBreed for breed of the litter (LLLL, YYLL, DDYL), Parity represented the parity of the sow at time of the recorded litter size (1, 2, 3, 4, ≥5), HerdYearSeason represented the field where the sow was housed during the birth of the litter and the lactation period (70 groups). Random effects were *IDsow* for the direct genetic effect of the sow, *PEsow* for the permanent environmental effect of the sow, and e for the residual effect. These random effects were assumed to be independent and followed normal distributions. *IDsow*~*N*[0,**A**σ_a_^2^], where σ_a_^2^ is the variance of *IDsow* and **A** is a relationship matrix based on pedigree information. *PEsow*~*N*[0,**I** σ_PE_^2^], where σ_PE_^2^ is the variance of *PEsow* and **I** is an identity matrix. *e*~*N*[0,**I** σ_e_^2^], where σ_e_^2^ is the variance of *e* and **I** is an identity matrix.

Based on the estimates for TNB and D11, phenotypic and genetic variance and covariance components for L11 were derived as:
(2)$$\sigma_{L11}^{2}=\sigma_{\mathrm{TNB}}^{2}+\sigma_{D11}^{2}-2\sigma_{\mathrm{TNB},\;D11}$$
(3)$$\sigma_{L11,\mathrm{TNB}}=\sigma_{\mathrm{TNB}}^{2}-\sigma_{\mathrm{TNB},D11}$$
(4)$$\sigma_{L11,D11}=-\sigma_{D11}^{2}+\sigma_{\mathrm{TNB},D11}$$
where σ^2^_TNB_ is the phenotypic or genetic variance of TNB, σ^2^_D11_ is the phenotypic or genetic variance of D11, and σ_TNB,D11_ is the phenotypic or genetic covariance between TNB and D11.

### 2.4. Breeding Schemes

This study aimed to improve L11 in organic pig populations by adjusting the breeding scheme. In organic pig populations, males are selected from external conventional populations. Females are selected on organic nucleus/multiplier farms [19]. Therefore, to improve L11 through adjusting the breeding scheme, the females selected at the nucleus/multiplier farms were the primary focus. An assessment of the optimal selection strategy for improving L11 was achieved by simulating breeding schemes that use different breeding goals for selecting females.

### 2.5. Breeding Goals

In the simulated breeding schemes, a breeding goal (BG) was used to select dams based on their estimated breeding value (EBV) for the aggregate genotype. This aggregate genotype was based on the additive genetic value of an individual for the trait(s) in the breeding goal. Two breeding goals were tested: (1) a single-trait breeding goal and (2) a two-trait breeding goal.
(5)$$\mathrm{Single}\;\mathrm{trait}\;\mathrm{BG}=v_{L11}L11$$
(6)$$\mathrm{Two}\;\mathrm{trait}\;\mathrm{BG}=v_{\mathrm{TNB}}\mathrm{TNB}+v_{D11}D11$$
where v_L11_, v_TNB_, and v_D11_ are the economic values (EVs) for L11, TNB, and D11, respectively. EV was not based on real economic inputs; rather, weights were given to traits in the breeding goal. The two-trait breeding goal fixed v_TNB_ to 1, but used a value for v_D11_ that ranged from −5 to 0 using steps of 0.05. In addition, one two-trait breeding goal was added, in which the EV for TNB was equal to 0 and D11 was equal to 1. In total, 102 breeding schemes that used a two-trait breeding goal were simulated.

The single-trait breeding goal was similar to the trait that, until recently, was used in pig breeding programs in Denmark [1,17]. Two-trait breeding goals represent an alternative approach for selecting for survival and litter size. A subset of breeding goals with EVs is presented in Table 2. Inbreeding in % per generation for the breeding goals in Table 2 can be found in Table A1.

### 2.6. Simulated Breeding Schemes

Simulations were performed with ADAM software [20], and breeding values were estimated using DMU [18]. Parameter inputs of the simulation were variance components estimated from model (1). These parameters were also used to predict breeding values for selection, as if the true variance components were known. Each breeding scheme used a different breeding goal for selecting females. The simulation of each breeding scheme was replicated 400 times.

For the simulation of a breeding scheme, individuals were placed in age groups based on their age in time steps, where each time step represented approximately six months. Hence, males or females in age group two were two time steps old (~1 year), and females in age group five were five time steps old (~2.5 years). Males and females were available for reproduction in age group two and two to five, respectively.

Breeding schemes were simulated for 30 time steps. Each time step, 30 males were randomly selected from the second age group and mated with 150 females. These 150 females originated from age groups two to five, and were truncation-selected based on their estimated breeding value (EBV) for traits in the breeding goal. Females had eight surviving offspring with phenotypes that had a 1:1 sex ratio. Approximately 600 female piglets and 600 male piglets were born each time step.

In each time step, sires and dams were selected using the following action steps. First, from the 600 newborn males, 30 sires were randomly selected. The remaining males were culled. This step modelled, in a simplified way, the use of sires from the conventional system, without controlling the selection of sires. Second, from the 600 newborn female piglets, 240 individuals were truncation-selected based on their EBVs to form age group one. This step mimicked the pre-selection of gilts for potential reproduction. Third, females were culled at random, mimicking the death or selling of dams. After random culling, 220 females remained in age group two, 190 in age group three, 140 in age group four, and 100 in age group five, totaling 650 females. All of these females had phenotypic records for TNB, D11 and L11. Each female had up to four repeated records. In step four, of these 650 females, 150 dams were truncation-selected based on their EBVs to reproduce offspring. At the end of each time step, all males from age group two and all females from age group five were culled.

### 2.7. Sensitivity Analysis

A sensitivity analysis was performed to test which factors contributed to differences in genetic gain for L11, when the single-trait breeding goal with only L11 was used or the two-trait breeding goal with TNB and D11 was used.

Table 3 presents an overview of the eight breeding schemes that were simulated for the sensitivity analysis. Four scenarios were tested, each of which had varied genetic correlation between TNB and D11, and heritability estimates for TNB and D11. Genetic correlation between TNB and D11 was either as shown in Table 1, 0.66 (original), strong (0.9) or weak (0.0). Heritability estimates for TNB and D11 were either as shown in Table 1, 0.09 and 0.06, respectively (original), or assumed to be equal and 0.09. The genetic variance of TNB and D11 was the same for all scenarios, whereas the residual variance of TNB and D11 was changed to adjust to each scenario. Each scenario was simulated twice: once for a breeding scheme using a single-trait breeding goal and once for a breeding scheme using a two-trait breeding goal. The single-trait breeding goal contained L11 only, and the two-trait breeding goal had an EV equal to 1 for TNB and −1 for D11. For each of the four scenarios, the annual genetic gains for L11 in breeding schemes using the single-trait breeding goal and two-trait breeding goal were compared.

### 2.8. Data Analyses

Preparation of input files for ADAM and DMU, and analyses of ADAM simulation output files were completed in R [21]. For each scenario, an output file from ADAM was generated that included the mean breeding value of the population for TNB and D11 per time step (30 time steps in total) based on 400 replicates. From the simulation outputs, the annual genetic gain was calculated as:
(7)$$\Delta G_{t}=2*\frac{\sum_{i=26}^{30}{\mathrm{BV}}_{t,i}-\sum_{i=10}^{13}{\mathrm{BV}}_{t,i}}{\sum_{i=26}^{30}i-\sum_{i=10}^{13}i}$$
where ∆G*_t_* is the annual genetic gain for trait *t* (L11, TNB, or D11), and BV*_t_*_,_*_i_* is the average breeding value for trait *t* among the offspring born in time step *i*. To calculate the annual genetic gain, we assumed two time steps per year, hence the multiplication by 2 to indicate the genetic trend per year.

For the sensitivity analysis, a *t*-test was used to test the significance of differences in the annual genetic gain for L11 between single-trait and two-trait selection.

## 3. Results

### 3.1. Simulations

Variance components for L11, TNB, and D11 are presented in Table 1. Mean TNB was 15.18 ± 4.04 (standard deviation) and mean D11 was 2.25 ± 2.41.

Figure 1 shows the annual genetic gain for L11, TNB, and D11 when different breeding goals were used. Genetic gain for L11 was similar for the single-trait and two-trait breeding goals when the economic value for D11 in the two-trait breeding goal was between −1.2 and −0.7.

Table 2 presents a detailed overview of the EVs for the trait(s), and the annual genetic gain for L11, TNB, and D11 in a subset of breeding goals. The annual genetic gain for L11 was significantly higher when a two-trait breeding goal with equal economic values for TNB and D11 was used (0.207), compared to a breeding goal with L11 only (0.201). When the two-trait breeding goal was used, the number of dead piglets declined (−0.003) when the economic value for D11 was −1.6 or lower.

### 3.2. Sensitivity Analysis

Table 3 shows the annual genetic gain for L11 for breeding schemes using four scenarios with different genetic correlations and heritability estimates for TNB and D11. The differences between the single-trait breeding goal and the two-trait breeding goal were significant when the heritability estimates for TNB and D11 were not equal, and when the genetic correlation between TNB and D11 was not weak.

## 4. Discussion

Our study shows that genetic gain could be maximized by separating a combined trait into two traits. We demonstrated this by comparing a single-trait breeding goal based on L11 to a two-trait breeding goal with equal weights for TNB and D11. This two-trait breeding goal yielded significantly more genetic gain for L11 compared to using the breeding goal with only L11 (Table 3). Of note, in practice, selection for the number of living piglets is completed on day 5 [10,17]. Using the number of living piglets on day 11 (L11) was considered acceptable, because the additive genetic correlation between L11 and living piglets at day 4 was close to unity (0.994; unpublished results, estimated from the data).

### 4.1. Opportunities of the Two-Trait Breeding Goal

Differences in genetic gain for breeding schemes using different breeding goals (Table 2) were caused by changes in the accuracies of EBV for sows. The sensitivity analysis showed that L11 was improved by accounting for differences in heritability for TNB and D11, and the non-zero genetic correlation between these traits (Table 3). While the difference in heritability for TNB (0.09) and D11 (0.06) was small, this difference showed that genetic gain for L11 was improved (0.207) compared to when heritability was assumed to be equal (0.201) (Table 3). As this difference in heritability increases, we expect that the benefit of separating the trait into two traits would also increase. Similarly, accounting for genetic correlations between two traits results in higher genetic gain for the separated trait. Our results support the selection index theory [22], whereby simultaneous selection for correlated traits with differences in heritability or economic importance is most effective when using multiple-trait index selection.

Using a multi-trait breeding goal allows for the economic value of traits to be adjusted according to the production system, country, population, and changing economic conditions [23]. Hence, separating traits could fine-tune breeding goals to facilitate direct genetic gain in a desired direction. However, the economic weighting should be adjusted with caution. If economic weights for TNB and D11 are not chosen appropriately, a negative trend for L11 arises (Figure 1).

### 4.2. Breeding for Lower Mortality in Organic Pig Production

The current study used piglet mortality in organic pigs as an example because of its relevance in organic production. Selection for fewer dead piglets per litter reflects the aims for improved welfare in organic farming [6,24]. Furthermore, data for TNB and D11 are routinely collected on organic pig farms, thereby facilitating the analyses. Ultimately, costs and complications could be minimized if this new selection strategy, enhancing survival, was to be implemented in organic breeding programs.

Current organic pig breeding in Denmark depends on semen from male pigs bred in the conventional system. Consequently, genetic gain in organic pigs would follow the genetic trend of the conventional pig population [19]. Conventional pig breeding programs in Denmark have, until recently, selected for better survival and increased litter size simultaneously through selecting for live piglets at day 5 [10,17]. Whether piglet mortality is reduced due to genetic selection or improved management, it has been declining for over a decade [10,11,25]. Yet, the current study showed that selecting for the number of live piglets is more efficient when using a multi-trait approach. From an organic farming perspective, using conventional genetics in combination with the recent single-trait approach is sub-optimal for genetically improving the population [19,22].

In the case of L11, a two-trait breeding goal would give breeders the option to include welfare directly in the breeding goal [13], and emphasize selection for fewer dead piglets. Early piglet survival is mostly determined by sow behavior and genetics [12,26,27]. Using the genetics of sows and the two-trait strategy with TNB and D11, an organic sow line focused on improved piglet survival could be established.

## 5. Conclusions

In conclusion, selection for a given trait could be optimized by separating it into two correlated traits with different heritability. Specifically, splitting L11 into TNB and D11 allowed for increased litter size, while reducing the number of dead piglets in a litter. This result is promising for organic pig breeding programs, which could improve piglet welfare without hindering improvements in litter size.

## Figures and Tables

**Figure 1 animals-12-01796-f001:**
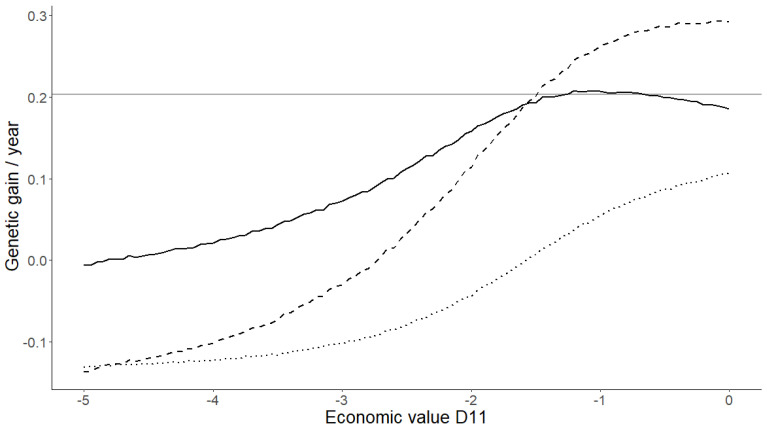
Annual genetic gain for living piglets on day 11 (L11), the total number born (TNB), and the number of dead piglets on day 11 (D11) for simulated breeding schemes with different breeding goals. Grey horizontal line shows the annual genetic gain for L11 when the breeding goal included L11 only. The three black lines show the annual genetic gain for individual traits when using a two-trait breeding goal with TNB and D11. Solid, dashed, and dotted lines indicate annual genetic gain for L11, TNB, and D11, respectively. For all two-trait breeding goals, the economic value was equal to 1 for TNB, and was between −5 and 0 for D11.

**Table 1 animals-12-01796-t001:** Heritability (diagonal), additive genetic correlation (below diagonal) and phenotypic correlation (above diagonal) estimates for the number of living piglets on day 11 (L11), the total number born (TNB), and the number of dead piglets on day 11 (D11). SDP represents the phenotypic standard deviation; Mean represents the means for the three traits. Standard errors of heritability estimates were ~0.02. Values for L11 were derived from TNB and D11 using Equations (2)–(4).

Trait	L11	TNB	D11	SDP	Mean
Alive at day 11	0.06	0.80	−0.18	3.33	12.92
Total number born	0.87	0.09	0.45	3.67	14.61
Dead at day 11	0.20	0.66	0.06	2.23	1.69

**Table 2 animals-12-01796-t002:** Annual genetic gain for the number of living piglets on day 11 (L11), the total number born (TNB), and the total number of dead piglets on day 11 (D11) under different breeding goals.

Breeding Goal	Economic Value			Annual Genetic Gain ^1^		
	L11	TNB	D11	L11	TNB	D11
Focus only on L11	1	-	-	0.201	0.227	0.026
Focus only on TNB	-	1	0	0.185	0.292	0.107
Focus only on D11	-	0	−1	−0.082	−0.223	−0.141
Equal focus on TNB and D11	-	1	−1	0.207	0.262	0.055
Emphasize D11 ^2^	-	1	−1.60	0.190	0.187	−0.003

^1^ SEs varied from 0.001 to 0.002; ^2^ breeding goal for which reduction in D11 is achieved.

**Table 3 animals-12-01796-t003:** Sensitivity analysis on the effect of differences in heritability or the strength of genetic correlation for the total number born (TNB) and dead piglets on day 11 (D11) on an annual genetic gain for live piglets at day 11 (L11). For each scenario, a single-trait breeding goal and a two-trait breeding goal were simulated.

Scenario		Annual Genetic Gain L11		
Genetic Correlation ^1^	Heritability	Single-Trait BG	Two-Trait BG	*p*-Value ^2^
original ^3^	original ^6^	0.201	0.207	>0.001
strong ^4^	Original	0.133	0.158	>0.001
weak ^5^	Original	0.356	0.353	0.236
original	equal ^7^	0.201	0.206	0.012

^1^ Additive genetic correlation (r_a_) between TNB and D11; ^2^ *p*-value from *t*-test comparing annual genetic gain for L11 for single-trait model and two-trait model; ^3^ original r_a_ = 0.66; ^4^ strong r_a_ = 0.9; ^5^ weak r_a_ = 0; ^6^ original h^2^_TNB_ = 0.09 and original h^2^_D11_ = 0.06; ^7^ equal h^2^ for TNB and D11, where h^2^_TNB_ = h^2^_D11_ = 0.09.

## Data Availability

Data supporting the reported results are provided and freely accessible.

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
