# Peer review of "Effective Selection for Lower Mortality in Organic Pigs through Selection for Total Number Born and Number of Dead Piglets"

_animals, 2022, doi:10.3390/ani12141796_

Round 1

Reviewer 1 Report

I have read the manuscript entitled “Selection for lower mortality in organic piglets is most effective when decomposing number the trait into total number born and number of dead piglets” by Zaalberg et al.

In terms of Animal welfare, lower number of piglets dead is promising, although the genetic correlation between TNB and D11 was estimated to be positive (Table 1). Zaalberg et al. have challenged to search a novel methodology to efficiently improve the number of piglets weaned under outdoor condition. They have tried to show the utility of selection based on both TNB and D11 rather than based on L11 (TNB-D11).

I this this study tackled important issue for efficient, healthy, and sustainable pork production. And I would like to propose some points to be improved, especially remove the obscurity ni Materials and Method, Results, and Discussion sections.

・Was “Breeding goals”, as the authors were denoted, actually the selection index? Please separately and clearly explain the definition of aggregate genotypes (breeding objective) and selection index in this study. Was the selection performed based on TBV or EBV?

・Were EBVs obtained via single-trait analysis for single trait BG and two-trait analysis for two trait BG? I suspect that the superiority of two-trait strategy in improving L11 was due to the difference in accuracy of EBV. Please show the results about the accuracy of EBVs. Also, was BLUP equation (or MME) solved using the value of genetic parameters, which were the same as those data simulation?

・I am interested in the results for the change in the inbreeding coefficient after selection among different schemes. Was using multiple traits benefit or harmful to keep the degree of inbreeding lower as much as possible?

・How about the applicability of restricted selection index especially in terms of lowering D11?

・Could you add the discussion about how to determine the appropriate weighting information because the results seems rather worse when applying inappropriate weights in two-trait selection scheme?

・P1L9: Affiliation No. 1 → No. 2

・P1L34: TMB → TNB

・P6L246-248: remove

Author Response

The comments to Reviewer 1 is uploaded as a file.

Reviewer 2 Report

      The first 5 days after birth is the peak period of death of piglets, most of the causes of death include low body weight, diarrhea, crushing, poor feeding, etc. Therefore, many breeders or pig producers advocate that the number of piglets alive 5 days after birth is an important indicator to measure the reproductive performance of sows.  Mortality between 5 and 11 days is very low and probably accounts for less than 5% of all lactation deaths.  Mortality within 5 days accounts for more than 90% of all lactation deaths.  The performance of piglets at 5 days may be more suitable for breeding and production.  I don't agree with the author that 11 days is more appropriate.  Whether the study documented correlations between performance data of piglets at 5 and 11 days of age. 

      In general, the coefficient of variation of reproductive performance such as litter size of commercial pig strains is generally less than 15%, and the degree of variation is appropriate, which reflects that the commercial pig strains have certain performance stability and certain coefficient of variation, which is conducive to the positive selection of traits.  The variation coefficient of D11 in Table 1 is only 5%, so it is debatable whether it is suitable for breeding. 

Author Response

The response to Reviewer 2 is uploaded in a file

Round 2

Reviewer 1 Report

The manuscript has been improved.

Author Response

Thanks for revising our manuscript. I am glad to see we have responded to your previous comments appropriately.

Reviewer 2 Report

  • The author answered the reviewers' questions better.

Author Response

Thanks for revising our manuscript. You indicated that the manuscript can be improved, but you have not given direct feedback. Therefore, I have not been able to improve the manuscript according to your current wishes. I hope the newly revised manuscript will be acceptable.